# Performance and Sociodemographic Determinants of Excess Outpatient Demand of Rural Residents in China: A Cross-Sectional Study

**DOI:** 10.3390/ijerph17165963

**Published:** 2020-08-17

**Authors:** Yanchen Liu, Yingchun Chen, Xueyan Cheng, Yan Zhang

**Affiliations:** 1School of Medicine and Health Management, Tongji Medical College, Huazhong University of Science and Technology, Wuhan 430030, China; ycliu97@hust.edu.cn (Y.L.); chenyingchun@hust.edu.cn (Y.C.); xycheng6972@hust.edu.cn (X.C.); 2Research Centre for Rural Health Service, Key Research Institute of Humanities & Social Sciences of Hubei Provincial Department of Education, Wuhan 430030, China

**Keywords:** excess healthcare demand, higher-level preference, hospital choice, health-seeking behavior, primary healthcare system, rural China

## Abstract

Excess healthcare utilization is rapidly rising in rural China. This study focused on excess outpatient demand (EOD) and aimed to measure its performance and sociodemographic determinants among China’s rural residents. A total of 1290 residents from four counties in central China were enrolled via multistage cluster random sampling. EOD is the condition in which the level of hospital a patient chooses is higher than the indicated level in the governmental guide. A multilevel logistic regression was used to examine the sociodemographic determinants of EOD. Residents with EOD accounted for 85.83%. The risk of EOD was 51.17% and value was 5.69. The value of EOD in diseases was higher than that in symptoms (t = −21.498, *p* < 0.001). Age (OR = 0.489), educational level (OR = 1.986) and hospital distance difference (OR = 0.259) were the main sociodemographic determinants of EOD. Excess outpatient demand was evident in rural China, but extreme conditions were rare. Results revealed that age, educational level and hospital distance were the main sociodemographic determinants of EOD. The capacity of primary healthcare institutions, universality of common disease judgement and understanding of institution’s scope of disease curing capabilities of residents should be improved to reduce EOD.

## 1. Introduction

Excess healthcare demand has been reported in numerous studies [1,2,3]. This entails that patients receive unnecessary healthcare services or those that are beyond their capacity to pay [4]. It includes the following forms: surging hospitalization rates, excess outpatient demand and inappropriate hospitalization [5]. Excess outpatient demand (EOD) is defined as the outpatient preference for an institution level that exceeds the patient’s reasonable need. Reasonable need can only be measured according to governmental guide [6,7] because China lacks general practitioners. EOD is identified when the level of hospital a patient chooses is higher than the indicated level in the governmental guide. For example, if a patient with a sore throat or tonsillitis chooses a township or county hospital even if only a treatment in a village clinic is needed, then this preference is considered EOD according to the guide.

China suffers from severe excess healthcare demand, such as inappropriate admission [8,9,10,11], but studies on EOD are few. In terms of system setup, the construction of China’s rural three-level healthcare delivery system is split with the development of marketization. Medical institutions compete to provide medical services in pursuit of high economic returns. On the patient side, residents without general practitioner’s guidance tend to go to high-level institutions under the policy of free choice of medicine to avoid the risk of misdiagnosis at the grassroots level, thereby driving EOD. We suspect that EOD may be prevalent in rural China, and the data collected in recent years support this conjecture [12]. From 2008 to 2018, the average annual growth rate of county hospital outpatients was 4.46%, whereas that of township hospitals and village clinics was merely 3.04% and 2.02%, respectively, during the same period.

EOD increases the financial burden of patients. The average annual growth rate of per capita health care expenditure for rural residents was 17.56% in 2008–2018 [12]. The inappropriate medical utilization is exacerbated; the rate of inappropriate admission in five county hospitals out of 2230 residents in central China was 15.2% in 2013 [10]. Congestion in high-level hospitals and the weakening or bankruptcy of primary hospitals contribute to the prominence of the contradiction between supply and demand [9]. EOD not only undermines primary care functions, but also crowds out resources for patients in need of high-level care. The fund’s pressure to pay Basic Medical Insurance for Urban and Rural Residents increases. Certain hazards at all levels make EOD the focus of China’s health reform.

The numerous negative effects of EOD—especially on common health—compel the examination of its prevalence. However, policies, a product of historical and social values, are hard to change [13]. Many reforms in free-access health care systems, including in France [14], Germany [15] and Belgium [16], did not lead to compulsory gatekeeping regulations, but rather resulted in stronger incentives for patients to reduce EOD with few effects [17]. Other countries that traditionally provide free choice to hospitals, such as China [18] and Japan [19], only piloted the gatekeeping function in small areas, but did not spread it widely. Considering these challenges, understanding EOD behavior can help in devising effective policies to reduce the behavior and improve the efficiency of health care delivery.

Current studies on excess healthcare demand focus on inpatient research and include inappropriate admissions [9,20,21], inappropriate inpatient services [11,22] and lack the outpatient perspective. EOD in most Western countries, where the general practitioner and guiding mechanism are mostly perfect, hardly exists. By contrast, EOD is a unique phenomenon in Asian countries, such as Korea [23], India [24], Japan [25] and China. Most studies on hospital choice pertain to countries with gatekeeping systems [26,27]. A few studies on EOD behavior, such as those conducted in the Netherlands [28], Japan [29], Israel [30], the United States [31,32] and African countries [33,34], analyzed factors or reasons associated with such behavior. The obtained or predicted bypass rates range from 13.7% in Japan [29] to 59% in Tanzania [34]. Therefore, EOD behavior is context dependent and that country-specific analyses are necessary to gain deeper information about the nature of EOD. Most studies in China only concentrated on the choice of medical institutions [35,36,37], and few quantitative investigations on the trend of EOD were conducted. Therefore, the issue of EOD in rural China is an important phenomenon that should be investigated. China has issued a tiered healthcare delivery system policy to cope with this trend, but its effect is not obvious because the mechanism and determinants of EOD are unclear. Moreover, the review of EOD in the medical insurance system urgently needs sociodemographic determinants.

Two internationally used methods for evaluating the appropriateness of excess healthcare demand are the recommendation method (implicit criteria) [38,39] and the indicator method (explicit diagnosis-specific criteria [40]. The former requires senior clinicians to review medical records and judge whether hospitalization is appropriate, and this decision depends entirely on the clinicians’ knowledge, experience and skills. Researchers are not required to provide any information [41]. The latter method provides reviewers with general criteria for all types of diagnosis, including disease severity, service intensity and the patient’s need for hospitalization [42]. The recommendation method is used by clinical medical specialists, whereas the indicator method is often employed by managers and other experts in nonclinical medicine. The results of the former method are more reliable than those of the latter. However, the recommendation method is also replete with various problems, such as limited funding and time of insurance companies, lack of authoritative experts and inconsistent judgment even for the same doctor at different times [42]. Therefore, the indicator method is more practical. The present research was to investigate the need and willingness of patients for healthcare demand instead of the behavior that has occurred. Thus, the recommendation method was not adopted herein. In summary, this study devised some self-designed indicators that represent the gap between the governmental guide and intended choice to measure EOD.

This study aimed to explore the status and sociodemographic determinants of EOD in rural China. This study offers advice for formulating better health care policies and promotes the effective use of medical services.

## 2. Materials and Methods

### 2.1. Study Setting

This study involved a cross-sectional survey focusing on adult rural residents whose main medical institutions of choice belong to three levels, namely, county, township and village. A self-designed questionnaire was used to investigate residents’ outpatient choice of institution level for five symptoms and seven diseases. For residents who chose self-treatment, the survey asked about the medical institutions where they buy drugs by themselves. The institution was treated as a village clinic if they chose nearby pharmacies, the medical level of which is equivalent to that of village clinics in China. The collection of basic patient information focused on sociodemographic factors, such as age, education, time required to reach the nearest county and township hospital and medical burden, to facilitate monitoring and prediction. Among these factors, the difference in distance between two hospitals of different levels was described by the time required to reach the institution via the most common mode of transportation. To represent the general medical needs of residents, we chose symptoms or names of 12 common diseases. The corresponding theoretical medical institutions devoted for these diseases were specified in the governmental guide, including two village–township, five township, four township–county and one county medical institution [6,7].

### 2.2. Data Sources

We conducted a cross-sectional survey from August to September in 2017 and adopted a multistage cluster random sampling method. First, we sampled from four provinces that are all located in the central plains of China and whose economic level is intermediate compared with that of other counties and cities in the country. We then selected counties with the third economic level in each province. The four counties selected, namely, Dingyuan, Xixian, Xialu and Renxian (Figure 1), are all close to the capital of the province or neighboring province, and the provincial hospitals can be reached in less than 2 h. Finally, we sampled in each county according to 0.4‰ of the population of each county; hence, we randomly selected two townships in four counties and then 3–4 villages in each township.

### 2.3. Data Processing and Statistical Analysis

We collected data via face-to-face inquiry. The quality of the questionnaire was guaranteed by face-to-face questioning and filling in by uniformly trained investigators. We collected a total of 1290 samples and all were effective. We entered the data in Microsoft Excel 2019 and established a database. A case represented a resident’s choice of institution for a disease or symptom.

We introduced a new self-designed indicator, namely, EOD value. This value refers to the difference between the residents’ choice of medical institution and that recommended in the governmental guide [6,7]. The value ranges from 1 to 5, which corresponds to village clinics, township hospitals, county hospitals, city hospitals and province hospitals, respectively. The EOD value of symptoms, diseases and overall situation was the sum of the value of five symptoms, seven diseases and their total, respectively. A resident with a positive total EOD value was identified as a resident with EOD. To supplement comparable indicators for research on the same subject, we used EOD risk. Excess outpatient demand risk refers to the rate of EOD; for example, the EOD risk of five symptoms is 1/5 if a resident has one positive EOD value of the symptom.

IBM SPSS Statistics 23.0 was used for statistical analysis. Correlation analysis was conducted on the EOD values of symptoms and diseases. *t*-test and ANOVA test were used to identify the characteristics of the samples, and a multilevel regression model was run on MLwiN 2.30 to examine the sociodemographic determinants of EOD.

### 2.4. Ethical Approval and Consent to Participate

The study protocol conformed with the guidelines of the Ethics Committee of the Tongji Medical College of Huazhong University of Science and Technology (IORG No: IORG0003571). Patient information was anonymized and de-identified before the analysis.

## 3. Results

### 3.1. The Level and Characteristic Distribution of EOD

As shown in Table 1, the samples’ average total EOD value was 5.69, indicating that in general, each patient had about five times to choose medical institutions higher by one level for the 12 cases listed. Excess outpatient demand value varied among counties. Xixian had the lowest total EOD value (3.39). Xialu had the least number of samples and the highest total EOD value (7.00). Renxian had the greatest number of samples and the second lowest total EOD value (6.21), similar to that of Dingyuan. The differences were statistically significant (*p* < 0.01).

We conducted one-way ANOVA with variables associated with the subgroups of each factor. With regard to individuals, younger, better educated or unmarried people and residents whose distance to county hospitals was closer than that to town hospitals were more likely to have higher EOD values than others in the same group (*p* < 0.001).

### 3.2. Excess Outpatient Demand of Symptoms, Diseases and Overall Situation

Residents with EOD accounted for 85.83% of the sample population. The proportion of polarized residents whose total EOD value was within the range of −8 to 6 or 12 to 14 was no more than 10% (Figure 2). The samples’ average total EOD risk was 51.17%, indicating that EOD behavior occurred 5 out of 10 times.

The mean and standard deviation of the EOD values of diseases were higher than those of symptoms (Table 2). After the paired-sample *t*-test, the means of the EOD value of diseases became higher than those of symptoms (t = −21.498, *p* < 0.001). However, a positive correlation was observed between the two; the higher the EOD value of symptoms was, the higher the EOD value of diseases was (R = 0.251, *p* < 0.001).

### 3.3. Samples’ Choice of Outpatient Institution in Five Symptoms and Seven Diseases

Table 3 summarizes the basic information of the samples’ choice of outpatient institution when five symptoms and seven diseases occurred. In general, when faced with the abovementioned symptoms or diseases, most residents displayed EOD behavior. The EOD values of each case other than those of A2, A4 and B3 were positive. In terms of diseases, pediatric-related diseases (such as childbirth), conditions that require surgery (such as acute appendicitis) and emergency situations (such as pesticide poisoning) were preferentially treated in county hospitals and had high EOD values. The EOD value of pneumonia cases among babies was negative possibly because the top-level institution in the governmental guide is a county hospital. A comparison of the choice of a disease and the choice of its symptom revealed that tonsil inflammation (A1 and B7) had similar EOD values. However, acute appendicitis (B5 and A4) had evidently higher EOD values when the disease was clear which may be related to the fact that people usually think that surgery is required as soon as the name of the disease appears.

### 3.4. Sociodemographic Determinants of EOD according to Multilevel Analyses

We performed a three-level logistic regression analysis at first. Individuals were assigned to level 1, townships were designated to level 2, and counties were indicated to level 3. However, only the variance of the null model of level 2 was statistically significant (χ^2^ = 230.20, *p* < 0.001), with aggregation of information at the township level.

Table 4 shows the outputs of final two-level (township–individual) analysis. The parameters estimated to be associated with random effects included variance in township and residuals. The between-township variability in EOD value was 23.4% (ICC = 0.234).

Individual variables were included in the multilevel analysis model. Results showed that the odds of samples reporting higher EOD values decreased with age. Compared with people who were well educated, residents with a lower education level were more likely to go to medical institutions of lower level. The shorter the time difference between households to county hospitals and township hospitals was, the higher the EOD value would be.

## 4. Discussion

### 4.1. Excess Outpatient Demand Behavior Was Evident, but with Rare Extreme Conditions in Rural China

Residents with EOD accounted for 85.83% of the total sample population. Considering the broad category of common diseases, EOD risk was 51.17%, which was remarkably higher than the 15% bypass rate associated with 52 minor conditions in Korea [23] and the bypass rate under the point-of-service plan in the United States [43].The overall situation of EOD was evident, but individual situations tended to differ. However, extreme cases were rare, and the choices facing all symptoms and diseases were higher or half of the choices were lower.

This discussion also considers our reflections during the conduct of field survey and the various perspectives affecting EOD found in the literature to obtain a comprehensive description of the mechanism of EOD. Excess outpatient demand is the result of the joint action of supply and demand. On the supply side, due to the fragmentation of China’s unique health service delivery system, the county hospitals, township hospitals and village clinics, which are the main choices for patients in rural areas, are no longer cooperative agencies that provide continuous care, but rather institutions that provide similar services separately [40,44]. However, the capacity of grassroots hospitals, especially township hospitals, is insufficient and far from satisfying their functions and lower than that of hospitals. The main cause of this deficiency is the inverted triangle system of resource allocation and lack of human and material resources [45,46,47]. Moreover, the unclear scope of services results in the lack of specificity [48]. Residents do not trust or know the level of treatment that grassroots hospitals can provide. On the demand side, under the policy of free choice of medicine, rural residents without a general practitioner choose hospitals on the basis of their own judgments about diseases and hospitals. In choosing a hospital, residents first rely on the perceived technical quality of healthcare professionals and then on the functional quality of attitude and convenience [49]. Patients believe that a hospital of higher level has a higher technical and functional quality and a better service capacity. To avoid the risk of misdiagnosis and treatment delay, residents who even live far from county hospitals choose higher-level institutions because of ease of transport. Moreover, past experiences have a lingering effect on residents’ visits [50]. A satisfying EOD visit result will lead to the next EOD.

### 4.2. Excess Outpatient Demand Was More Evident When Residents Knew the Exact Disease rather than the Symptoms

The mean and median of the EOD values of diseases were 2.31 and 3 points higher than those of symptoms. This finding showed that the EOD phenomenon of knowing the exact diseases was more evident than that of when only symptoms were detected. The *t*-test also confirmed this result.

Residents who chose high-level institutions when symptoms appear were more likely to choose higher-level institutions when they knew the exact disease, especially in pediatric-related diseases (such as childbirth), conditions that require surgery (such as acute appendicitis) and emergency situations (such as pesticide poisoning). This finding may be related to the limitation of the knowledge of rural residents about the disease and the professional inducibility of the name of the disease, which is due to the residents’ own judgment of their condition. As the saying goes, “a long illness makes a doctor”. During the visits, residents presenting various symptoms that hinted to common diseases, such as tonsil inflammation, became familiar with the diseases and learned that they were not serious. Hence, the EOD behavior was not obvious.

A plausible reason for EOD behavior in pediatric-related diseases is that Chinese parents value their children and are more sensitive to children’s diseases than conditions that afflict adults. Situations that require surgery may be considered as mild according to symptoms that initially appear to be “normal”, such as stomach pain. When the name of the disease was specific, the residents’ perceptions of early surgical treatment were elicited. Thus, the disease was considered serious. Research shows that when residents have health problems, they judge themselves according to their existing health knowledge, network health information and feelings. Based on this premise, two simple and effective medical judgment frameworks were constructed, and the patients chose medical institutions accordingly. The results of previous studies were consistent with the findings that residents tended to choose primary medical institutions or community health facilities when the illness was mild, and they were inclined to choose medical institutions or general hospitals at the county level or above when the illness was chronic and serious [51,52]. Excess outpatient demand behavior in emergency situations may occur because people require treatments that may lead to rapid recovery. Hence, they consider a hospital with better technology.

### 4.3. Age, Educational Background and Distance to Hospital Were the Main Sociodemographic Determinants of EOD

People over the age of 60 years tended to have less EOD because they consider the hospital background and the presence of expert doctors. Young people’s modern concept of medical comfort leads to their choice of high-quality diagnosis, treatment, environment and attitude, and hence, “superior” hospitals. The primary concern of patients with chronic diseases was convenience and the cost of medical treatment. The daily medical treatment plan is relatively mature, and many stay in the village. Therefore, many patients in outpatient clinics in village-level medical institutions have obvious choice preferences [53,54].

Residents with higher education levels were more likely to have high EOD than those with a lower educational background. This trend may be due to the supposition that people with a high education level have a high understanding of medical service information. Thus, they have a higher emphasis on and investment in health than people with a lower education level. Accordingly, under the permission of China’s policy of free choice of medicine, people with better education—who usually have higher incomes—are more inclined to choose nonprimary medical institutions to receive better medical services and cured sooner despite the higher, but affordable medical costs [55], rather than spend more energy to primary institutions and risk getting misdiagnosed.

Residents who required more than 30 minutes to go to the county hospital than to the township hospital tend to have less EOD. This result emphasized that the accessibility of the institution leads to high EOD. The proximity of an institution is a favorable factor that contributed to the residents’ choice of going to higher-level institutions for half of the diseases. If the residents deem that the county hospital is too far from them compared to the township hospitals, then they may opt to avail of services from primary medical institutions because of their lower income, advanced age, less disposable time and inconvenient mobility. They may do so even if they have insufficient confidence in the quality of care given by these primary medical institutions. They lack the means to go to large hospitals; thus, they choose to go to primary health care institutions to seek medical care [56].

### 4.4. Limitations of This Study

This study has several limitations. First, this study was a cross-sectional study because the time allotted for the investigation and the places that could be sampled were limited. Hence, the conclusions may not be comprehensive. Second, the self-design nature of indices related to EOD lacks scientific basis and validation. Thus, the EOD description may not be accurate. Third, the theoretical institution guide followed herein was from Henan Province and may not be suitable for other provinces and counties. The counties sampled herein have not yet issued relevant guidelines. Fourth, the scope of diseases was limited. The 12 common diseases selected in this study may not be quite correct. If there is a classification study, then the EOD values may be different. Fifth, this study only focused on sociodemographic factors. Moreover, the influencing factors were not sufficiently explored.

Specifically, the indicator “EOD value” introduced herein has certain limitations. First, the theoretical institutions in the guidelines may not be applicable to other provinces or countries, and they may change as institutions develop. Second, residents may choose multiple institutions for outpatient care at the same time and what we take here is the first medical institution with the greatest possibility. Third, residents’ choice is based on their preference for treatments when a certain symptom is presented or a certain disease is diagnosed; however, a gap exists between the preference and the actual behavior. Therefore, this research is applicable only under certain conditions. A similar research should further adjust the indicators or introduce other more general indicators, such as EOD rate.

## 5. Conclusions

This study quantitatively analyzed the performance and sociodemographic determinants of EOD. Data were obtained from 1290 residents from four regions in central China. Given the limitations of the study, the following hypotheses can be raised that the phenomenon of EOD was evident in China. The EOD of specific diseases was more likely to become higher than that of symptoms. Older, less educated residents, and people who live far from county hospitals tended to have lower EOD values.

For residents, information and education campaigns should be conducted so that certain behaviors concerning seeking medical help can be corrected. Sufficient health information, about common diseases and scope of disease curing capabilities of institutions at all levels, will help residents better view their own diseases, prevent residents from treating common diseases as serious ones and underestimating symptoms severity and then have it matched to the proper medical institutions. Of course, it is undeniable that some other sociological factors influence here, such as the availability of information channels such as the media and the Internet, the understandability of the language of publicity information, the availability of testing tools or regularly voluntary consultation by the institutions, etc. Moreover, the behavior of seeking inappropriate medical treatment on the basis of other factors (multisubject and multichannel) must be addressed by various means, such as by strengthening the supervision and verification of service effectiveness and reforming the government compensation and third-party payment methods for medical insurance. These actions can be implemented to rationalize the reimbursement gap between lower- and higher-level institutions, alleviate the economic burden of residents suffering from various diseases, enhance the capacity of primary medical care and link primary medical care with government medical insurance. These positive reinforcements will encourage residents to seek medical treatments at the grassroots level first. A two-way referral system should be established to improve the efficiency of medical service utilization, implement the grading of diagnoses and treatment concepts and promote the healthy development of the medical system.

The results of the present study has several research applications. At the theoretical level, the self-defined EOD indicator introduced herein offers new research perspectives. At the practical level, the discovery of EOD phenomenon and the analysis of its causes further suggest that institutional guidelines should be adjusted, and the direction of the tiered healthcare delivery system policy should be improved.

We suggest that future research should expand the scope of diseases to investigate and explore general evaluation indicators or adopt the indicator “EOD rate” to allow comparison of results. Moreover, it would be better to choose a place where the local authority such as health administrative department has published the official guidelines of services in institutions at all level to conduct a preference survey. Further research should design a novel method for exploring the potential of the indicator and recommendation method of evaluation. Finally, the sociological and demand side factors of EOD behavior can be analyzed to understand better the mechanism of EOD.

## Figures and Tables

**Figure 1 ijerph-17-05963-f001:**
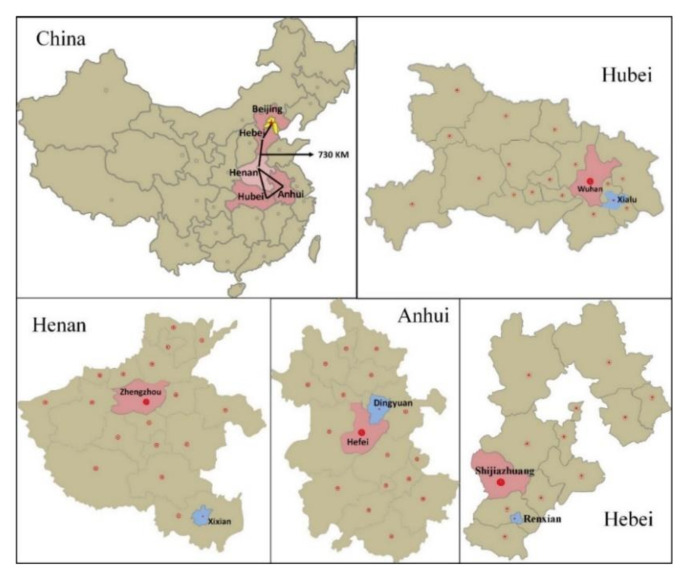
Sampling sites.

**Figure 2 ijerph-17-05963-f002:**
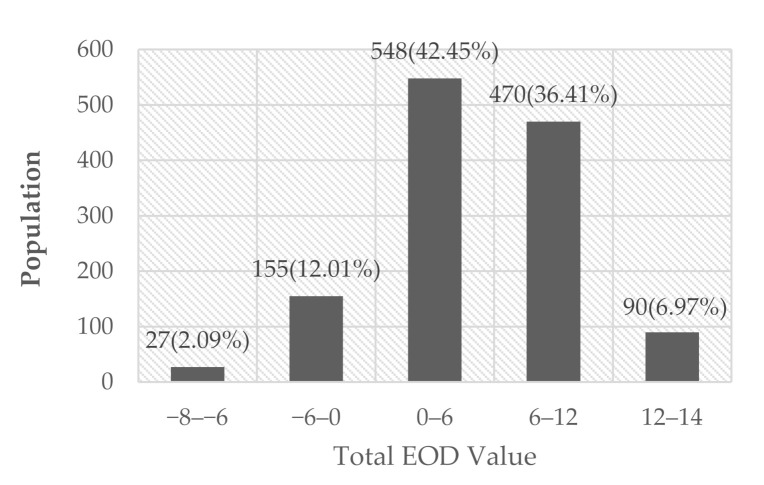
Distribution of total excess outpatient demand (EOD) value.

**Table 1 ijerph-17-05963-t001:** Characteristics of the samples (*n*,%; *n* = 1290).

Sociodemographic Factors	*n* (%)	Total EOD Value	t/F
All	1290 (100)	5.69 ± 4.97	
County			
Dingyuan	256 (19.84)	6.29 ± 4.27	32.277 **
Xixian	306 (23.72)	3.39 ± 5.26	
Xialu	222 (17.22)	7.00 ± 3.70	
Renxian	506 (39.22)	6.21 ± 5.18	
Sex			
Male	587 (45.50)	5.57 ± 4.81	0.640
Female	703 (54.50)	5.79 ± 5.11	
Age ^1^			
18–39	296 (22.95)	7.38 ± 4.46	23.962 *
40–59	542 (42.02)	5.38 ± 5.01	
≥60	452 (35.04)	4.95 ± 5.00	
Education			
Lower than PS ^2^	590 (45.74)	4.39 ± 5.05	34.465 *
JHS ^3^	392 (30.39)	6.10 ± 4.67	
SHS ^4^	226 (17.52)	7.26 ± 4.48	
Above College	82 (6.36)	8.74 ± 4.30	
Marital status			
Married	1166 (90.39)	5.61 ± 4.97	8.572 *
Unmarried	59 (4.57)	8.17 ± 3.78	
Divorced/widowed	65 (5.04)	4.82 ± 5.40	
Economic burden			
Severe	88 (6.82)	4.83 ± 4.48	1.467
Moderate	372 (28.84)	5.68 ± 4.83	
Not at all	830 (64.34)	5.78 ± 5.08	
Time difference (min) ^5^			
<0	142 (11.01)	8.49 ± 5.12	24.939 *
0–15	267 (20.70)	6.26 ± 4.57	
16–30	392 (30.39)	5.62 ± 4.59	
>30	489 (37.91)	4.63 ± 5.09	

^1^ The interviewed residents are all adults (≥18 years old in China) who are considered to be able to judge independently; ^2^ PS—primary school; ^3^ JHS—junior high school; ^4^ SHS—senior high school; ^5^ Time difference; single time difference between households to the nearest county hospital and township hospital; * Indicates significance at the 0.001 level (*p* < 0.001); ** Indicates significance at the 0.01 level (*p* < 0.01).

**Table 2 ijerph-17-05963-t002:** EOD value and risk of symptoms, diseases and total situation (*n* = 1290).

	Total		Symptoms		Diseases	
	EOD Value	EOD Risk (%)	EOD Value	EOD Risk (%)	EOD Value	EOD Risk (%)
Mean ± SD	5.69 ± 4.97	51.17 ± 22.20	1.69 ± 2.96	42.84 ± 30.59	4.00 ± 3.32	57.01 ± 24.56
Min − Max	−8–24	0–100	−3–12	0–100	−5–17	0–100
Median	6	50	2	40	5	71

**Table 3 ijerph-17-05963-t003:** Residents’ medical willingness for each symptom or disease (*n*,%; *n* = 1290).

Symptoms ^1^/Diseases	Village Clinics	Township Hospitals	County Hospitals	EOD Value
A1 Swollen and sore throat	721 (55.89)	466 (36.12)	103 (7.98)	0.52 ± 0.64
A2 Long-time upper abdominal pain	533 (41.32)	418 (32.40)	339 (26.28)	−0.14 ± 0.82
A3 Cough extending for 2 years	492 (38.14)	339 (26.28)	459 (35.58)	0.99 ± 0.88
A4 Pain in the lower right abdomen	579 (44.88)	488 (37.83)	223 (17.29)	−0.27 ± 0.75
A5 Diarrhea, watery and blood in stool	168 (13.02)	288 (22.33)	834 (64.65)	0.60 ± 0.81
B1 Childbirth	1 (0.08)	110 (8.53)	1179 (91.39)	1.08 ± 0.52
B2 Mild pneumonia in adults	241 (18.68)	389 (30.16)	660 (51.16)	0.39 ± 0.87
B3 Baby pneumonia	102 (7.91)	209 (16.20)	979 (75.89)	−0.21 ± 0.75
B4 Disc herniation	121 (9.38)	369 (28.60)	800 (62.02)	0.67 ± 0.86
B5 Acute appendicitis	39 (3.02)	286 (22.17)	965 (74.81)	0.83 ± 0.67
B6 Pesticide poisoning	75 (5.81)	364 (28.22)	851 (65.97)	0.73 ± 0.79
B7 Tonsil inflammation	710 (55.04)	496 (38.45)	84 (6.51)	0.52 ± 0.62

^1^ The full description of five symptoms is in the Appendix A.

**Table 4 ijerph-17-05963-t004:** Multilevel analysis of sociodemographic determinants of EOD (*n* = 1290).

Sociodemographic Determinants	Coef.	*p*	95% CI	OR
Fixed part:				
Intercept	18.130	<0.001	16.209, 20.052	74,775,436.5
Sex				
Male	Ref.			
Female	0.219	0.163	−0.089, 0.528	1.245
Age ^1^				
18–39	Ref.			
40–59	−0.489	0.027	−0.923, −0.054	0.613
≥60	−0.715	**0.005 ****	**−1.212, −0.217**	**0.489**
Education				
Lower than PS ^2^	Ref.			
JHS ^3^	0.423	0.035	0.030, 0.815	1.527
SHS ^4^	0.686	0.007 **	**0.189, 1.183**	**1.986**
Above college	1.183	**0.001 ***	0.461, 1.906	0.489
Marital status				
Married	Ref.			
Unmarried	0.632	0.105	−0.133, 1.396	1.881
Divorced/widowed	0.251	0.486	−0.456, 0.959	1.285
Economic burden				
Severe	Ref.			
Moderate	0.357	0.272	−0.280, 0.994	1.429
Not at all	0.412	0.181	−0.192, 1.016	1.510
Time difference (min) ^5^				
<0	Ref.			
0–15	−0.678	0.018	−1.238, −0.117	0.508
16–30	−0.775	0.004 **	−1.309, −0.243	0.461
>30	−1.351	**<0.001 ***	**−1.877, −0.826**	**0.259**
Random-effect Parameters:				
Township	–	–	6.477, 19.086	–
Residual	–	–	19.324, 20.879	–

^1^ The interviewed residents are all adults (≥18 years old in China) who are considered to be able to judge independently; ^2^ PS—primary school; ^3^ JHS—junior high school; ^4^ SHS—senior high school; ^5^ Time difference; single time difference between households to the nearest county hospital and township hospital; * Indicates significance at the 0.001 level (*p* < 0.001); ** Indicates significance at the 0.01 level (*p* < 0.01); the bold numbers: on the line with significance at the 0.001 or 0.01 level.

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
