# Peer review of "Performance and Sociodemographic Determinants of Excess Outpatient Demand of Rural Residents in China: A Cross-Sectional Study"

_ijerph, 2020, doi:10.3390/ijerph17165963_

Round 1

Reviewer 1 Report

A very interesting study under China's health reform. Some modifications need to be done: 

  1. Needs extensive language editing. Sometimes I found myself in difficult to follow the meanings.
  2. The phase of "Excess healthcare demand" or "Excess outpatient demand". This expression can easily lead to ambiguity, it seems that the authors want to express unreasonable utilization beyond the guideline.
  3. it's unclear of the sampling method. If the author considered the income level and the number of residents in each county.
  4. should also present how many samples selected, not only effective samples.
  5. it's unclear how the authors handle with self-treatment individuals.
  6. From table 3, it seems that the ANOVA test and t-test were used, but in the methods, the authors said t-test and chi-square test.
  7. table 3 should be presented first.
  8. More * usually stands for lower p-value. inversed in this study.
  9. it's unclear how the multilevel logistic regression was conducted, and there is no null model in table 4.
  10. lack a discussion of the limitation the indicator used in this study.

Reviewer 2 Report

  • 35-36: patients' needs do not necessarily correspond to the governmental guide, a difference might be made between both;
  • 43: EOD can be induced by medical institutions as much as by patients themselves. You might elaborate a bit on this point, to show that you're aware of both phenomena, and provide objective and positive reasons for focusing on the patient side. This is particularly important since you claim to take sociological factors into consideration.
  • 84-90: See 35-36. In addition, the recommendation method is an important determinant of EOD. Stating that "it's not practical" is not enough to exclude this method. Instead, you might clearly assume that you decided to refer to the indicator method only, and provide positive reasons for that decision. Also mention in the conclusion that further research are needed to explore the potential of the recommendation method.
  • in general: you should make it clear why you decided to focus on the patient side only. Indeed, focusing on medical institutions could lead to interesting findings, particularly in terms of policy recommendations.
  • 99: what you describe as "sociological factors" are in my view sociodemographic indicators. Sociological factors correspond, for example, to representations the residents have of different types of institutions (care services). You should use the right terms and mention sociological factors among the perspectives for future research, which not mentioned in the conclusion.
  • 214-221: new elements such as the patients' judgment on disease, hospitals as well as their past experiences are referred to. But they are neither mentioned before nor taken into account in the research design. Then, you should explain that you are going to discuss your results based on the literature and provide justification for which type of literature you refer to. In this respect, the claims you make based on the literature are much more sociological than the factors included in your research design.
  • 256-261: you might consider the opposite claim. Indeed, higher education and a better knowledge of the healthcare system could also result in a better utilisation of the system, which would entail, according to international standards, to refer to primary care before hospitals. 
  • 260: you seem to assume that the quality of care is lower in primary care than in hospitals. If you state that, you should provide explanation for that statement. Moreover, is quality of care the relevant difference between both? Couldn't you consider their different roles or functions in the health care system? and shouldn't you mention this difference earlier?
  • 287: be careful that the statement you make about publicity and education is somehow contradictory with the claim you previously made (256).
  • I regret that no perspective for further research are provided.

Round 2

Reviewer 2 Report

the conclusion, as it is formulated in the current version, is not really supported by the results. The section concerning the limitation of the study mentions that:

  1. "Therefore, this research is applicable only under specific conditions and simply provides new research perspectives

Then, in the conclusion, the authors state that:

  1. "The results demonstrated that the phenomenon of EOD was evident in China"

In my view the problem can be addressed by formulating the conclusion a bit differently, for example, by mentioning that, given the limitations of the study, the following hypotheses can be raised to explain...then, the authors might provide perspectives for further research which would enable to overcome the limitations and test the hypotheses. This would improve the overall consistency of the article.

In addition, I advise the author to be more careful when they put forward that improving health literacy is "THE" solution. Indeed, several studies showed that improving health literacy was efficient but not sufficient to "help residents rationally monitoring their health status...". Other factors (among other sociological factors) influence people behavior in this respect. Please, just make it clear you're aware of that.

Concerning English language, this version is easier to read. So it's fine for me but cannot say if this meets the editor's expectations.

Finally, the terms (sociological-socio-demographic) have not been adapted in the abstract.
